# The Effects of Swallowing Disorders and Oral Malformations on Nutritional Status in Children with Cerebral Palsy

**DOI:** 10.3390/nu14173658

**Published:** 2022-09-04

**Authors:** Mustapha Mouilly, Adil El Midaoui, Aboubaker El Hessni

**Affiliations:** 1Departement of Biology, Faculty of Sciences and Techniques, Errachidia, Moulay Ismail University of Meknès, BP. 509 Boutalamine, Errachidia 52000, Morocco; 2Departement of Pharmacology and Physiology, Faculty of Medecine, University of Montreal, Montreal, QC 999040, Canada; 3Departement of Biology, Faculté des Sciences, Ibn Tofail University of Kénitra, Kénitra 14000, Morocco

**Keywords:** nutrition, cerebral palsy, swallowing, oral malformations, Morocco

## Abstract

Nutrition plays an important role both from a nutrition and a socio-psychological point of view; this part seems to be even more crucial in cerebral palsy where undernutrition is responsible for an increase in morbidity and mortality. The objective of this study was to evaluate the effects of swallowing disorders and oral malformations on the nutritional status of children with cerebral palsy. We evaluated 65 patients aged 2 to 17 years using a cross-sectional, descriptive and observational approach. All patients had a definite diagnosis of cerebral palsy. The measurement of anthropometric variables (weight, height, Body Mass Index (BMI) and circumferences) was performed according to recognized techniques and measurements. The Z-score was also calculated using the World Health Organization (WHO) references. The 5-level Gross Motor Function Classification System was used, providing a standardized classification of motor disability patterns for children with cerebral palsy. The population had a median age of 9.25 (4.50–16.00) and was about 53% female. Furthermore, 75% of the patients had a height inferior to 158 cm. The results of our study show that 42 (64.6%) had false routes, 17 (26.2%) had oral-facial malformations and 51 (78.5%) did not have lip prehensions during meals. The results also show that growth retardation is closely related to gross motor function with *p* = 0.01, as well as all nutritional indices (Z-score weight for age, Z-score height for age and Z-score BMI for age) are affected by swallowing disorders and oral malformations, with statistically significant values < 0.05. In conclusion, a preventive and curative management specific to this population of children with cerebral palsy must be implemented with an interdisciplinary concertation.

## 1. Introduction

Cerebral palsy (CP) is the most common physical disability in children, affecting 3.6/1000 children in three areas of the United States in 2002 [1] and is between 1.5 and 4 per 1000 children worldwide [2]. CP corresponds to: a group of permanent developmental disorders of movement and posture, resulting in activity limitation, that are attributed to non-progressive disturbances occurring in the developing fetal or infant brain. Motor disorders in CP are often accompanied by disturbances in sensation, perception, cognition, communication, behavior, epilepsy and secondary musculoskeletal problems [3]. The motor function is known to be the most common factor in cerebral palsy, but growth and nutritional disorders should not be overlooked. 

Children with CP frequently present with feeding barriers that can alter their nutritional status [4]. Dysphagia, gastroesophageal reflux disease (GERD) and gastrointestinal disorders such as constipation are the main barriers to feeding in children with CP [4]. In addition, these children may also present with hypospadias or xerostomia and poor hydration, related to the presence of dysphagia or a lack of communication of thirst [4]. This can lead to malnutrition, the development of deficiencies in certain nutrients and osteopenia, but also to a delay in weight gain [4,5]. In addition, overweight is present in 10–15% of children with neurological disorders [4].

CP affects not only gross motor functions, but also can have an impact on swallowing and feeding. Children with CP are at increased risk of growth failure and malnutrition [6,7]. This can have repercussions on the quality of life of the patient [4] and also of those accompanying him or her [8]. In a recent study, findings from the Global LMIC CP Register show that most children with CP had undernutrition and older age, low maternal education and spastic tri/quadriplegia, and Gross Motor Functional Classification System (GMFCS) levels III–V were significant predictors of underweight [9]. Motor ability according to a global classification system should also be considered in the management of children with CP. Few studies have investigated the relationship between oromotor dysfunction and nutritional indices.

The objective of our study is to evaluate the effects of swallowing disorders and oral malformations on the nutritional status of children with cerebral palsy and to compare levels of gross motor function versus nutritional indices in this population.

## 2. Materials and Methods

### 2.1. Participants

The study was conducted in a rehabilitation center in northwest of Morocco with outpatients. It was explained to the parents or guardians that participation in the survey was optional and that the data would be collected anonymously. Informed consent was signed once agreement was positive.

We evaluated 65 patients aged 2 to 17 years using a cross-sectional, descriptive, observational approach. All patients had a definitive diagnosis of cerebral palsy, established by the prescribing physician. Patients using a nasogastric feeding tube and those with genetic diseases were excluded. Patients complaining of fever, pneumonia, diarrhea or surgical complication during the evaluation were excluded. General information, such as sex, age and etiology of cerebral palsy, was obtained from medical records.

### 2.2. Measurement

Measurement of anthropometric variables was performed using the following techniques and measurements. Weight: measured by the difference between the weight of the caregivers and the total weight of the children on the caregivers’ lap (due to the patients’ inability to stand). A Filizola^®^ digital scale (Filizola, São Paulo, Brazil) was calibrated to zero for use. Knee height: using a Cescorf^®^ caliper (Cescorf, Porto Alegre, Brazil) and with the child’s leg at a 90° angle, measure the length from the anterior surface of the leg to the sole of the foot. Height estimation: method used for patients unable to maintain an orthostatic position for direct height measurement. We used the knee value and applied it to specific mathematical formulas [10,11]. Arm Circumference (AC): the child’s arm was placed in a flexed position. The midpoint between the acromion and the olecranon was marked with a non-elastic measuring tape. Thus, the arm circumference was measured with the limb standing at the marked point.

Patients’ weight, height and BMI were assessed according to the reference curves of Brooks et al.; (2011) [12]. As in the original study. Referring also to the guidelines published by Mehta et al. (2009) [13], those defined as nutrition risk had a BMI-for-age index below the 25th percentile. The Z-score was also calculated using the World Health Organization (WHO) references. Circumference values were measured.

The Gross Motor Function Classification System (GMFCS) is a 5-level system providing a standardized classification of motor disability patterns for children with cerebral palsy from birth to 18 years of age. Distinctions between the 5 levels, ranging from Level I (less limitation) to Level V (greater limitation), are based on limitations in motor function and mobility device needs in four age ranges. The GMFCS has good reliability and validity [14,15] Figure 1.

### 2.3. Statistical Methods

Data were processed using Excel software. The open-ended questions and the answers were coded. Regarding the statistical analysis and the elaboration of the output tables, the treatments were performed using the free statistical software (SPSS) version 13.0. The normality of the quantitative variables was tested. The quantitative data following the normal distribution were expressed in mean standard deviation if not in median (Q1–Q3) and the qualitative ones in percentage. The correlations between the two parameters were estimated by the Pearson or Spearman correlation coefficient according to normality. The means of the parameters were compared using parametric tests for comparison of means (Student *t*-test). Comparisons of percentages were made by Chi2 tests or Fisher exact tests. For all tests performed, the risk of permissible error was set at α = 0.05.

## 3. Results

### 3.1. Sociodemographic and Clinical Data

Our study included 65 patients with cerebral palsy. The age ranged from 2 years to 17 years with a median of 9.25 (4.50–16.00). The weight in our population was a median of 37 (22–49). In total, 75% of the patients had a height inferior to 158 cm. The female sex predominated with a frequency of 52.3%. Perinatal etiology was the most frequent with 44.6%. Level IV and V of the GMFCS were the most represented with respectively 33.8% and 36.9%. Bronchial superinfections were present in 40% of the population with a frequency of twice a year in 60%. It should also be noted that all of our population was fed orally, even for patients with GMFCS Level V. In total, 27.7% of our population is under epileptic treatment, while 32.3% take neuroleptics Table 1.

### 3.2. Characteristics of Swallowing Disorders and Oral Health Status

In the results of our study, it was found that the swallowing disorders of children with cerebral palsy are presented in the form of false routes in 64.6% and particularly when drinking, gastroesophageal reflux was present in 7.7%. Constipation as an intestinal transit disorder was found in 52.3% of the population and around 95% of the population have a nutritional mode in the form of texture, but only 6% of the population need to gel water when hydrating. In the results of this study, only 1.5% of the population using a laxative treatment. The salivation during feeding was noted in the majority of the population (98.5%). It was also noted that 78.5% of our population had difficulties with lip grip during feeding. Nauseous reflexes were found in 4.6% of the population. The textured feeding mode was adopted in 95.4% of the population and 72.3% of our population had difficulties in feeding themselves during meals Table 2.

Regarding the oral status of the patients in our population, 17 (26.2%) had oral-facial malformations. The status of the dentition was incorrect in 20%, and the retraction of the upper lip was found in 56.9% of the population. The gingival bleeding was noted in 9.2% of the population. The result of this study showed that 90.8% of the population had no bad breath. It was also noted that the majority of the patients in our population had no preferential side when taking food (Table 3).

### 3.3. Nutritional Indices

According to the GMFCS, it was found that Level V had more stunting with a statistically significant value *p* = 0.01; otherwise, concerning wasting and underweight variables, they did not show any statistical significance between the different levels of the GMFCS Table 4.

The results of our study show highly significant differences regarding nutritional indices versus oral malformations and swallowing disorders. Regarding the weight-for-age z-score, the difference between patients with oral-facial malformations and those without is significant with *p* = 0.001; otherwise, regarding false routes, a significant difference was recorded *p* = 0.01. The height-for-age z-score also showed a statistically significant difference regarding oral-facial malformation and false routes with *p* = 0.01 and *p* = 0.0001, respectively. The difference was also significant when evaluating the nutritional index of emaciation (BMI z-score for age) versus the existence of malformations, on the one hand, and the suffering of false routes, on the other hand, with *p* = 0.03 and *p* = 0.0001, respectively (Table 5).

## 4. Discussion

The results of this study showed that children with cerebral palsy in this population have a median age of 9.25 (4.50–16), a median weight of 37 (22–49), a median height of 146 (121–158) and a median BMI of 15.90 (13.93–19.10). This examination is the most important part of the physical examination, the results reported by Day et al. (2007) in their comparative study show that the weight, height and BMI of cerebral palsy patients fed by gastrostomy tube feeding (GTF) were higher than those fed orally. [16] Other studies also show that these anthropometric parameters are significantly below norms in children with neurodevelopmental disabilities [17,18,19,20]. Although most children with cerebral palsy are undernourished [21], our results show that a significant part of our population is under antiepileptic and neuroleptic treatment; it has been noted in the literature that anticonvulsant medication can impair appetite and, consequently, growth. [22]. In addition, very often children with cerebral palsy are unable to feed themselves according to our study. Strauss et al. (1998) suggested that children fed by GTF had a higher risk of death than those who had a possibility to feed themselves and to be fed orally (risk ratio = 23.65) [23].

Our results highlighted the existence of swallowing disorders and oromotor dysfunction in the population studied. This factor has been described in other studies as very frequent [24] and presents a major risk of malnutrition [5,25,26,27]. Dysfunctional swallowing makes oral feeding difficult and is often associated with inadequate caloric intake [28,29]. The gastroesophageal reflux has been reported to be present in our population and very common in children with CP in other studies [30,31] and even increases nutrient losses through subsequent vomiting, and reflux esophagitis can also cause discomfort that prompts the child to refuse to feed [22]. In addition, swallowing dysfunction can cause gastroesophageal reflux pulmonary aspiration, which is common in our population in the form of chronic respiratory disorders, recurrent pneumonia and respiratory symptoms suggestive of chronic aspiration. Readjustments of the consistency of the food in texture were adopted to best suit the child. 

In this study, oral malformations are frequent, this was also observed by D.Hurtel et al. in adults with sequelae of infantile cerebral palsy causing consequent difficulties in mastication and swallowing [32].

The results of our study show that the most affected children show more growth retardation; this was also reported in another study [18], which showed that the growth of children not only with spastic quadriplegia level IV and V of the GMFCS is more affected, but also in children with diplegia or hemiplegia.

The results of this study show highly significant differences in nutritional indices versus oral malformations and swallowing disorders; studies have also found a very high risk of malnutrition closely related to oromotor dysfunction [5,26]. Therefore, patients must be detected as early as possible in order to implement a strategy that is likely to be effective, such as enteral feeding. Indeed, oral disorders have functional repercussions and increase swallowing disorders and finally disturb nutrition. The study by D. Hurtel et al. suggests prevention, early detection and individualized treatment of oral pathologies in a multidisciplinary concertation to break this pathological cycle [32].

Despite considerable effort, this study had few limitations, such as its small sample size and being a single-center study, but it can be a starting point for other studies on nutritional biological parameters in the population of children with cerebral palsy.

## 5. Conclusions

To evaluate the effects of swallowing disorders and oral malformations on the nutritional status of cerebral palsy children, our study allowed us to highlight the frequency of swallowing disorders and oral malformations in this population. The results of this study show that the most severely affected children according to the GMFCS present more growth delay. Highly significant values for nutritional indices affected by oral malformations and swallowing disorders were found.

These nutritional indices in the face of oromotor dysfunction and oral malformations addressed in this study will need to be investigated in a broader setting in combination with thorough biological examinations and long-term follow-up.

To minimize the effects of oromotor dysfunction and oral malformations on the nutritional status of children with cerebral palsy, multidisciplinary work with parental involvement should be implemented. It must ensure prevention, early detection and treatment of oral malformations in an individualized manner.

## Figures and Tables

**Figure 1 nutrients-14-03658-f001:**
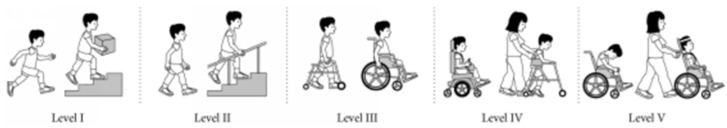
Global Motor Function Classification System (GMFCS).

**Table 1 nutrients-14-03658-t001:** Sociodemographic and clinical data of patients.

Characteristics	Value N = 65
Age (years) **	9.25 (4.50–16.00)
Weight (Kg) **	37 (22–49)
Height (cm) **	146 (121–158)
Sex ***	
Male	31 (47.7)
Female	34 (52.3)
Age group ***	
2 to 7 years old	36 (55.4)
8 to 11 years old	5 (7.7)
12 to 17 years old	24 (36.9)
BMI class ***	
Less than 13	5 (7.5)
Between 13 and 18.5	33 (49.3)
Between 18.5 and 20	13 (19.4)
Between 20 and 25	16 (23.9)
Brachial circumference *	17.95 ± 4.54
Calf circumference *	20.75 ± 5.08
Nature of the disability ***	
Spastic	42 (64.6)
Dyskinetic	15 (23.1)
Hypotonic	8 (12.3)
Convulsion ***	
Yes	20 (30.8)
No	45 (69.2)
Anti-epileptic treatment ***	
Yes	18 (27.7)
No	47 (72.3)
Neuroleptic treatment ***	
Yes	21 (32.3)
No	44 (67.7)
Bronchial superinfection ***	
Yes	26 (40)
No	39 (60)
Number of pneumopathies per year ***	
One per year	8 (12.3)
Two per year	39 (60)
Four per year	11 (16.9)
Five per year	7 (10.8)
Etiologies ***	
Prenatal	19 (29.2)
Perinatal	29 (44.6)
Postnatal	12 (18.5)
Unknown	5 (7.7)
GMFCS ***	
Level I	7 (10.8)
Level II	5 (7.7)
Level III	7 (10.8)
Level IV	22 (33.8)
Level V	24 (36.9)
Sibling Rank *** (N = 65)	
First	30 (46.15)
Second	17 (26.15)
Third	12 (18.46)
Fourth	5 (7.7)
Fifth	1 (1.54)

* Mean ± SD; ** Median (Range). *** Effective (%).

**Table 2 nutrients-14-03658-t002:** Characteristics of swallowing disorders in cerebral palsied patients.

Characteristics	Value (N = 65)
False routes	
Yes	42 (64.6)
No	23 (35.4)
False routes while eating	
Yes	11 (16.9)
No	54 (83.1)
False routes while drinking	
Yes	42 (64.6)
No	23 (35.4)
Gastroesophageal reflux	
Yes	5 (7.7)
No	60 (92.3)
Transit problems	
Constipation	34 (52.3)
Diarrhea	18 (27.7)
Alternation	13 (20)
Laxative treatment	
Yes	1 (1.5)
No	64 (98.5)
Autonomy at mealtime	
Yes	18 (27.7)
No	47 (72.3)
Nutritional mode	
Texture	62 (95.4)
Diet	3 (4.6)
Hydration	
Water	61 (93.8)
Gelled water	4 (6.2)
Lying down during the meal	
Yes	4 (6.2)
No	61 (93.8)
Seating during the meal	
Yes	60 (92.3)
No	5 (7.7)
Nausea reflex	
Yes	3 (4.6)
No	62 (95.4)
Lip prehension during feeding	
Yes	14 (21.5)
No	51 (78.5)
Salivation during feeding	
Yes	64 (98.5)
No	1 (1.5)

**Table 3 nutrients-14-03658-t003:** Oral health status characteristics of cerebral palsied patients.

Characteristics	Value (N = 65)
Oral-facial malformation	
Yes	17 (26.2)
No	48 (73.8)
Condition of the teeth	
Correct	52 (80)
Incorrect	13 (20)
Gingival bleeding	
Yes	6 (9.2)
No	59 (90.8)
Upper lip	
Retracted	37 (56.9)
Not retracted	28 (43.1)
Lip prehension during feeding	
Yes	7 (10.8)
No	58 (89.2)
Bad Breath	
Yes	6 (9.2)
No	59 (90.8)
Preferred side	
Right	5 (7.7)
Left	6 (9.2)
None	54 (83.1)

**Table 4 nutrients-14-03658-t004:** Nutritional indices in relation to patients’ gross motor ability levels.

	Global Motor Function Classification System (GMFCS)						
	Level I	Level II	Level III	Level IV	Level V	Fisher’s Exact Test	*p* Value
Emaciation (BMI-for-age, z-score < −2 SD)							
Yes	1 (4)	0 (0)	2 (8)	10 (40)	12 (48)	6.493	0.16
No	6 (15)	5 (12.5)	5 (12.5)	12 (30)	12 (30)		
Growth retardation (height-for-age, z-score < −2 SD)							
Yes	1 (4.8)	1 (4.8)	0 (0)	5 (23.8)	14 (66.7)	11.977	0.01
No	6 (13.6)	4 (9.1)	7 (15.9)	17 (38.6)	10 (22.7)		
Underweight (weight-for-age, z-score < −2 SD)							
Yes	0 (0)	1 (4.5)	2 (9.1)	7 (31.8)	12 (54.5)	4.175	0.25
No	0 (0)	3 (17.6)	4 (23.5)	5 (29.4)	5 (29.4)		

**Table 5 nutrients-14-03658-t005:** Nutritional indices according to oral-facial malformation and swallowing disorders.

	Z-Score Weight for Age				Z-Score Height for Age				Z-Score BMI for Age			
	N	Mean ± SD	t	*p* Value	N	Mean ± SD	t	*p* Value	N	Mean ± SD	t	*p* Value
Oral-facial malformation												
No	48	−1.239	2.685	0.001	48	−1.037	2.848	0.01	28	−2.088	2.326	0.03
Yes	17	−3.259			17	−2.983			11	−3.854		
False routes												
No	12	−1.239	2.685	0.01	23	−0.485	3.46	0.001	23	−0.576	4.117	0.0001
Yes	27	−3.185			42	−2.127			42	−2.421		

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
