# Peer review of "The Effects of Swallowing Disorders and Oral Malformations on Nutritional Status in Children with Cerebral Palsy"

_nutrients, 2022, doi:10.3390/nu14173658_

Round 1

Reviewer 1 Report

Title of study: No comments

Abstract: I did not understand these lines” In our population 50% of the . . . . . female sex predominates with 52.4%”

Introduction: The first sentence of first paragraph “Cerebral palsy (CP) is the most common physical disability in children, affecting 3.6/1000 children” need to be specify. You need to write either in Morocco or worldwide?

Remove the words “According to Rosenbaum et al,”

This sentence seems incomplete “Although the primary value loss in CP is motor function, growth and 41 nutrition disorders are common” Rephrase it.

Remove the sentence “Nutrition plays an important role in our lives, both from a dietetic and a socio-psychological point of view; this part seems even more crucial in children with cerebral palsy where undernutrition is responsible for an increase in morbidity and mortality, in the average length of hospitalization and consequently, in its cost [3-4], hence the need for an early and adapted evaluation and management.

Connect first sentence of second with third paragraph first sentence, then continue with second paragraph second sentence. Moreover, the third sentence of third paragraph need to be part of second paragraph, but you need to merge the content of third sentence of third paragraph with existing first sentence of third paragraph.

“Children with CP frequently present with feeding barriers that can alter their nutritional status. Dysphagia, gastroesophageal reflux disease (GERD) and gastrointestinal disorders such as constipation are the main barriers to feeding in children with CP. In addition, these children may also present with hyposialia and poor hydration, related to the presence of dysphagia or a lack of communication of thirst [5].” Then continue rest of the content of second paragraph.

Also increase number of citations.

The last paragraph of introduction section need to be revised: Remove the prevalence of GERD and other disorders, if you want to present prevalence of different types of disorders type from X to X %. Moreover, specify the population.

CP is one type of neurological disorder, so it could not be use as an analogous term to neurological disorder. Rework on your last paragraph of introduction.

Methodology:

Add subheading in the methodology section. Follow STROBE guidelines for cross-sectional observational studies for reporting of methodology.

Use word categorical instead of qualitative, because the data of your study is quantitative, which has two major types of data: (1) Numerical and (2) Categorical.

Results: Add some headings within your result sections. I found some of the interpretations in your results, you can write only those things, which you observed from your study. Rather than writing your own thoughts and also removed some references. Additionally you need to add full stop or point in rather than using comma (use . rather than ,), This comma (,) make me confused in several stance.

Discussion: nicely written

Add limitations of study based on your study methods and findings.

Author Response

Response to Reviewer 1 Comments

Abstract

Point 1: I did not understand these lines” In our population 50% of the . . . . . female sex predominates with 52.4%”

Response 1: The two quantitative variables age and height do not obey to the normal distribution; we have expressed them by the median and the quartiles (50% equivalent to the median and 75% to Q3). However, we must put a full stop just after, to address female predominance. This clarification was mentioned in Line 22 of the revised manuscript.

Introduction:

Point 2: The first sentence of first paragraph “Cerebral palsy (CP) is the most common physical disability in children, affecting 3.6/1000 children” need to be specify. You need to write either in Morocco or worldwide?

Response 2:  affecting 3.6/1000 children in three areas of the United States in 2002. According to the literature, the prevalence of CP is between 1.5 and 4 per 1000 worldwide. The reference has been added. Lines 35-36 of the revised manuscript.

Point 3: Remove the words “According to Rosenbaum et al,”

Response 3: As suggested by the reviewer. The modification has been made.

Point 4: This sentence seems incomplete “Although the primary value loss in CP is motor function, growth and 41 nutrition disorders are common” Rephrase it.

Response 4: As suggested by the reviewer, this sentence has been changed. The Motor function is known to be the most common factor in cerebral palsy; however, growth and nutritional disorders should not be neglected. Lines 41-42

Point 5: Remove the sentence “Nutrition plays an important role in our lives, both from a dietetic and a socio-psychological point of view; this part seems even more crucial in children with cerebral palsy where undernutrition is responsible for an increase in morbidity and mortality, in the average length of hospitalization and consequently, in its cost [3-4], hence the need for an early and adapted evaluation and management.

Response 5: As suggested by the reviewer. The modification has been made in the revised manuscript.

Point 6: Connect first sentence of second with third paragraph first sentence, then continue with second paragraph second sentence. Moreover, the third sentence of third paragraph need to be part of second paragraph, but you need to merge the content of third sentence of third paragraph with existing first sentence of third paragraph.

“Children with CP frequently present with feeding barriers that can alter their nutritional status. Dysphagia, gastroesophageal reflux disease (GERD) and gastrointestinal disorders such as constipation are the main barriers to feeding in children with CP. In addition, these children may also present with hyposialia and poor hydration, related to the presence of dysphagia or a lack of communication of thirst [5].” Then continue rest of the content of second paragraph.

Response 6: As suggested by the reviewer. The modification has been made in the revised manuscript.

Point 7: Also increase number of citations.

Response 7: As suggested by the reviewer. The modification has been made.

Point 8: The last paragraph of introduction section need to be revised: Remove the prevalence of GERD and other disorders, if you want to present prevalence of different types of disorders type from X to X %. Moreover, specify the population.

Response 8: As suggested by the reviewer. The modification has been made (Section revised)

Point 9: CP is one type of neurological disorder, so it could not be use as an analogous term to neurological disorder. Rework on your last paragraph of introduction.

Response 9: As suggested by the reviewer. The modification has been made (We have revised this paragraph and we have focused our research on cerebral palsy data)

Methodology:

Point 10: Add subheading in the methodology section. Follow STROBE guidelines for cross-sectional observational studies for reporting of methodology.

Response 10: As suggested by the reviewer. The modification has been made (subheading in the methodology has been added)

Point 11: Use word categorical instead of qualitative, because the data of your study is quantitative, which has two major types of data: (1) Numerical and (2) Categorical.

Response 11: We have used qualitative terms to describe technics of measurement and the methodology.

Results

Point 12: Add some headings within your result sections. I found some of the interpretations in your results, you can write only those things, which you observed from your study. Rather than writing your own thoughts and also removed some references. Additionally you need to add full stop or point in rather than using comma (use . rather than ,), This comma (,) make me confused in several stance.

Response 12: As suggested by the reviewer.The modification has been made. Please note that the values in square brackets in the results section do not represent references but are just the median (Range).

Point 13: Add limitations of study based on your study methods and findings.

Response 13: As suggested by the reviewer. The modification has been made in the revised manuscript in lines 206-207.

Reviewer 2 Report

Mouilly et al has done a study aimed to evaluate the effects of swallowing disorders and oral malformations on the nutritional status of children with cerebral palsy. The study has shown that most severely affected cerebral palsy children according to the GMFCS present more growth delay. Highly significant values for nutritional indices affected by oral malformations and swallowing disorders were found. Although this study has a few limitations, such as its small sample size, single-center, it can be one of the other studies analyzing effects of swallowing disorders and oral malformations on the nutritional status of children with cerebral palsy.

The abstract is written well, and it reflects the facts in the article. Materials/methods and results are well elaborated, and relevant references are cited. However, I suggest correcting a few grammar errors and few suggestions listed below. The article deserves publication after minor corrections.

Page 2 Line 53 - correct spelling "Hypospadias"

Page 3 line 121- can the author explain what false route means?

Page 3 line 122- please correct the decimal point. “78.5%”

Page 5 Table 1 - correct the spelling “Value”

Page 8 line 154 - Median weight and median length data is misleading for a reader to understand. The author can consider including the age range/median age in the discussion along with weight and length. Also the author can consider mentioning Median BMI which may be more helpful for overall understanding.

Page 9 line 180 and 192: please use et al.

Author Response

Response to Reviewer 2 Comments

Point 1: Page 2 Line 53 - correct spelling "Hypospadias"

Response 1: As suggested by the reviewer. The modification has been made. Line 46.

Point 2: Page 3 line 121- can the author explain what false route means?

Response 2:  False route means swallowed the wrong way = false food routes.

Point 3: Page 3 line 122- please correct the decimal point. “78.5%”

Response 3: As suggested by the reviewer. The modification has been made. Line 130.

Point 4: Page 5 Table 1 - correct the spelling “Value”

Response 4: As suggested by the reviewer. The modification has been made.

Point 5: Page 8 line 154 - Median weight and median length data is misleading for a reader to understand. The author can consider including the age range/median age in the discussion along with weight and length. Also the author can consider mentioning Median BMI which may be more helpful for overall understanding.

Response 5: As suggested by the reviewer. The modification has been made.

Point 6: Page 9 line 180 and 192: please use et al.

Response 6: As suggested by the reviewer. The modification has been made.

Reviewer 3 Report

This novel study is a cross sectional evaluation of the associations among swallowing disorders, oral malformations, and gross motor disabilities in children with CP and nutrition status, as indicated by BMI z scores.

Because this population is under-studied, and in particular studies on nutrition status and meaning of low weight for age are lacking, I believe this study has strong merit. There are several significant problems with the manuscript, but I think this may relate in part to custom. With editing assistance, I think it can be revised to make it more clear and complete.

Abstract:

line 12: prefer replacing term "dietetic" with "nutrition"

line 15: avoid using the term "cerebral palsied children". Person-first language would suggest using children with cerebral palsy, as is used elsewhere.

lines 23-25. Revise to include the mean or median age with SD, include the weight (not the height in cm): The population had a mean age of ____ +/- ____, and were about 53% female.

Introduction:

line 43: replace "dietetic" with "nutrition"

line 53: xerostomia is a more common word in the US. Maybe this term is more common internationally.

Methods:

lines 67-68. Revise to say whether this was conducted according to Helsinki guidelines. "The study was conducted in an outpatient rehabilitation center in northwest Morocco. Parents or guardians of children were invited to participate. Informed consent was obtained . . ."

lines 89-90. These don't meet the current guidelines for pediatric malnutrition. Also, current recommendations suggest avoiding use of disease specific growth charts in children. If this is used, I would also recommend using the WHO z scores for assessing malnutrition and the guidelines set forward in Mehta et al or Becker. Another approach would be to say that those children with BMI for age <25%ile were "at nutrition risk" rather than malnourished.

Results:

As in the abstract, revise the participant data: see notes for abstract.

line 123: "gag reflex?"

Table (1?) Replace "Valeur" with English: Median (Range). 

Tables 2-3: include more description than "value"

Discussion:

line 153: person-first language "children with CP".

In general, I think this study provides an important update to literature on children with CP. The tables seem a little simplistic. Several could be combined to make the key tables have greater impact. In addition, gross motor skills are not really mentioned in the abstract, but they are an important part of the analysis, so they should be mentioned in the purpose.

With significant revision, I think this manuscript should be accepted for publication.

Author Response

Response to Reviewer 3 Comments

Abstract:

Point 1: line 12: prefer replacing term "dietetic" with "nutrition"

Response 1: As suggested by the reviewer. The modification has been made.

Point 2: line 15: avoid using the term "cerebral palsied children". Person-first language would suggest using children with cerebral palsy, as is used elsewhere.

Response 2: As suggested by the reviewer. The modification has been made.

Point 3: lines 23-25. Revise to include the mean or median age with SD, include the weight (not the height in cm): The population had a mean age of ____ +/- ____, and were about 53% female.

Response 3: As suggested by the reviewer. The modification has been made. Please note that the two variables age and height are of abnormal distribution, we have expressed them by the median and the IQR, with an interpretation of 50% for age and 75% for height. Lines 22-23.

Introduction:

Point 4: line 43: replace "dietetic" with "nutrition"

Response 4: As suggested by the reviewer. The modification has been made. This section has been revised.

Point 5: line 53: xerostomia is a more common word in the US. Maybe this term is more common internationally.

Response 5: As suggested by the reviewer. The modification has been made. Line 46

Methods:

Point 6: lines 67-68. Revise to say whether this was conducted according to Helsinki guidelines. "The study was conducted in an outpatient rehabilitation center in northwest Morocco. Parents or guardians of children were invited to participate. Informed consent was obtained . . ."

Response 6: Yes indeed, our research had the objective to evaluate the effects of swallowing disorders and oral malformations on the nutritional status of children with cerebral palsy… Parents or guardians of children with cerebral palsy were invited to participate voluntarily with informed consent and respect of anonymity with the authorization of the center's managers. The general principles of the Helinski declaration were adopted.

Point 7: lines 89-90. These don't meet the current guidelines for pediatric malnutrition. Also, current recommendations suggest avoiding use of disease specific growth charts in children. If this is used, I would also recommend using the WHO z scores for assessing malnutrition and the guidelines set forward in Mehta et al or Becker. Another approach would be to say that those children with BMI for age <25%ile were "at nutrition risk" rather than malnourished.

Response 7: We would like to thank you for this very interesting remark. In fact, given the particularity of children with cerebral palsy, we have focused on the works published in this field, such as Brooks et al. and the references of the WHO and Perence L. et al. And also considering that as of today, there is still no consensus on the best definition of pediatric malnutrition according to Andrea MacCarthy et al. and to our knowledge that there are no nutritional guidelines for this vulnerable group of population, we have adopted the term malnutrition as described by Brooks et al. But referring to your comment and the guidelines defined by Mehta et al. in the nutritional support of the critically ill child. The change has been made. Lines 89-90

Results:

Point 8: As in the abstract, revise the participant data: see notes for abstract.

Response 8: As suggested by the reviewer. The modification has been made.

Point 9: line 123: "gag reflex?"

Response 9: A gag reflex occurs in the back of the mouth and is triggered when the body wants to protect itself from swallowing something foreign. This is a natural response, but it can be problematic if it’s overly sensitive.

Point 10: Table (1?) Replace "Valeur" with English: Median (Range). 

Response 10: As suggested by the reviewer. The modification has been made. The title of table 1 was above.

Point 11: Tables 2-3: include more description than "value"

Response 11: As suggested by the reviewer. The modification has been made. Descriptions have been added in paragraph 2 and 3 in the results section of the revised manuscript.

Discussion:

Point 12: line 153: person-first language "children with CP".

Response 12: As suggested by the reviewer. The modification has been made.